# Detection of Additives with the Help of Discrete Geometrical Invariants

**Raoul Nigmatullin [1],\*, Artem Vorobev [1] , Herman Budnikov [2], Artem Sidelnikov [3] and Elza Maksyutova [4]**

[1] Radioelectronics and Informative Measurements Technics Department, Kazan National Research Technical University (KNRTU-KAI), Karl Marx Str. 10, 420111 Kazan, Russia; vartems14@gmail.com

[2] Institute of Chemistry, Kazan Federal University, Kremlyovskaya Str. 18, 420008 Kazan, Russia; herman.budnikov@kpfu.ru

[3] Technology Department, Ufa State Petroleum Technological University (USPTU), Kosmonavtov Str. 1, 450000 Ufa, Russia; artsid2000@gmail.com

[4] Chemistry Department, Bashkir State University (BSU), Z. Validy Str. 32, 450076 Ufa, Russia; elzesha@gmail.com

\* Correspondence: rrnigmatullin@kai.ru

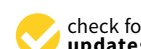

**Featured Application: The proposed methodology outlined in this work will be useful for detection of small additives in different complex fluids. It has a wide region of applicability in electrochemistry, chromatography etc., where different spectroscopic methods for detection of substance "traces" are used.**

**Abstract:** In this paper, we propose a general mathematical method for the detection of electrochemical additives in a given solute with the help of discrete geometrical invariants (DGI). This idea is based on the generalization of Pythagor's theorem that can be proved for two random sets located in the two-dimensional (2D) plane. This statement follows from the previous ideas proposed by Babenko, who essentially modernized the well-known theorem and propagated it on a wide class of "right" discrete sets with different symmetry. However, attentive analysis of these results shows that there is a possibility for their further generalization. For practical purposes, it is important to have discrete and deterministic curve(s) with the limited number of parameters that enables comparing two *random* sets of *any* nature if their quantitative description expressed in terms of the "best-fit" model is *absent*. Under the best-fit model, we imply the microscopic model that enables describing the measured data in terms of the minimal set of the fitting parameters. We propose at least two invariants: (a) the curve of the second order that coincides with the classical ellipse oriented at an arbitrary direction relative to the *X*- and *Y*-axes, and (b) the curve of the fourth order that has eight quantitative parameters and includes the cross-combination of the integer moments. In this paper, the DGIs of both types were used. These curves are made useful for the solution of a key problem in electrochemistry, i.e., the detection of small concentrations of D-tryptophan ($6.54 \div 38.7$) $10^{-5}$ mol$\cdot$L$^{-1}$ in a given solute (phosphate buffer solution ($Na_2HPO_4 + KH_2PO_4$) with pH = 6.86) that was activated by electrodes of two types—Pt (platinum) and C (carbon). The DGI method is free from treatment errors and model suppositions; therefore, it can be applied for the detection of small additives in a given solute and a further description can be attained with the help of a monotone/calibration curve expressed by means of parameters associated with the DGI.

**Keywords:** small-additive detection; electrochemistry; discrete geometrical invariants; D-tryptophan; C and Pt electrodes

## 1. Introduction and Formulation of the Problem

Under electrochemical methods of analysis, we imply the decoding and description of measured voltammograms (VAGs) associated with interphase boundary solute/electrode interactions of a multicomponent solution, containing a set of electroactive compounds. The desired VAGs give a possibility to extract qualitative/quantitative information about the content of the substances that participate in electrochemical redox reactions on the electrodes. The modern instrumental possibilities of volt-amperometric methods are characterized by high sensitivity and expressivity in the detection of different organic/non-organic substances. Thanks to this possibility, the measured VAGs (we imply the dependencies of $J(U)$) are widely used for the detection of trace quantities in different solutes and substances. In the last few decades, different numerical methods were widely used for the treatment of measured data in electrochemistry. They allow treating multidimensional data including a large number of data points. These methods also accelerate also express analysis for the detection of many substances that are present in multicomponent solutes. In theory and practice of analytical chemistry, this progress leads to the appearance of computer-oriented analytical systems for the simultaneous detection of substances having the same nature and the classification of complex substances, establishing their similarity/difference using chemometric methods. Special attention in modern publications was paid to the problem of detection and quantitative evaluation of the presence of *trace* substances. It is known that different analytical methods [1–3] allow determining the content of different analytes in a relatively wide range of small concentrations with acceptable accuracy. However, for achieving this aim, a researcher uses the preliminary results of their separation in order to avoid distortions of this analysis, or a researcher uses specific methods of accumulation of the desired analytical signal. The complex content of the initial signal requires the usage of more sensitive and selective methods for proper detection. The idea of their unification with highly sensitive methods should be supported by new methods of mathematical modeling, whereby the quick development of nanoelectronics is unified with computerized methods of treatment to determine the physical and chemical parameters accompanying these signals. We observe the tendency to miniaturization of existing analytical systems having all the necessary characteristics of macro-devices: (a) "on-a-chip" laboratories [4,5], (b) "electronic tongue" and "electronic nose" type systems [6,7], and (c) nano sensory "smart" detectors [8,9]. The social challenge, reflecting the population's needs in terms of increasing the quality of food, water, and medicine, is concentrated on the creation of compact, reliable, and sensitive pocket analyzers. The quality and reliability of these pocket devices/gadgets depends on some important factors. One can select them based on random fluctuations of a weak signal (noise) and its strong dispersion covering the range of working analyte concentrations. Without special mathematical tools and other methods, it becomes impossible to keep the desired concentration in the required range. Therefore, the problem of evaluation of the fluctuations, as well as their "quantitative reading", control, noise clipping, and, in some cases, noise gaining, remains an actual problem in electrochemistry. For the finding of diagnostic and mathematical methods, it is necessary to develop rather "general" approaches that should be sensitive, reliable, relatively simple (from the mathematical point of view), and free from treatment errors, in addition to (if it is possible, from model assumptions) implementation as an portable device.

In this paper, we present the results of voltammetric research of a temporal series of current and potential, formed in repeated redox processes of solute components in conditions where one electroactive analyte is slightly varied. As the analyte to be varied, we chose the D-tryptophan amino acid. This choice is supported by the fact that this acid participates in different electrochemical oxidation reactions, varying its activity depending on the material of the used electrode. It is important also that D-tryptophan refers to one enantiomer, and its presence as a mixture in pharmaceuticals is extremely undesirable. Therefore, one can expect its signal on a level comparable with that of random fluctuations/noises. The application of a multisensor electrochemical system becomes inefficient if the number of detected components is large and the sensitivity of each micro-sensor forming the desired system is low. The selection of D-tryptophan as an analyte is stipulated also by the

fact that, depending on the material of the electrode (C or Pt), this amino acid exhibits different electric activity in the oxidation reaction. At "trace" concentrations of D-tryptophan, both electrodes "feel" the concentration variations in the presence of a microcomponent. This behavior of the used multisensor having low sensitivity is characterized as "delayed" behavior. In many cases, this behavior is related to the mathematical method used for the extraction of a weak signal [10–13]. In our case, on C-electrodes, the D-tryptophan is oxidized easily with formation of a clearly expressed signal; however, on Pt-electrodes, it does not give clearly oxidized peaks. Therefore, part of the useful information is lost.

In this paper, we propose a general method free from treatment errors, which can improve the sensitivity of existing multisensor systems. In this paper, we try to solve the following problems:

(1)　Determination of some output parameters that are related to monotone concentration of the electroactive components in conditions of strongly fluctuating currents on the C("active") and Pt ("inertial") electrodes, for construction of the desired calibration curves.

(2)　Comparison of the obtained characteristics for the C- and Pt-electrodes and derivation of the calibration/monotone curves with respect to the given D-tryptophan concentration.

For solution of these problems, we propose a general method that enables comparing two random sets in a two-dimensional (2D) plane. The advantages of the proposed method are as follows:

(a)　Its generality (universality) implies the potential to compare *any* couple of random sets assigned by their coordinates $(Y1_k, Y2_k)$ in a 2D plane.

(b)　All parameters are expressed in terms of the integer moments and their inter-correlations.

(c)　The number of parameters is finite, and they have clear geometrical interpretation.

(d)　The proposed method oriented for analysis of discrete sets is free from *any* unjustified supposition and treatment errors.

(e)　The proposed approach can compare random sequences having different sampling volumes in the frame of the *same* set of statistical parameters.

Here, it is appropriate to compare the DGI approach with classical statistical mechanics, when a small number of statistical parameters associated with their averaged motion (T, *V*, *P*, μ) replaces a large number of parameters (3*N*) characterizing the micromotions of atoms and molecules. In our case, two random sequences having 2*N* data points are reduced to the analysis of eight statistical parameters associated with integer moments and their inter-correlations. In addition to this reduction, one can see also the desired curve in a 2D plane (Equations (A7) and (A18)), reflecting the behavior of these parameters. The proposed approach is free from model assumptions related to the nature of the compared random sequence/noise. It gives the unique possibility to establish some correspondence between conventional model noises (such as Gaussian, "white", "color", and stationary/non-stationary noises) and the deterministic curves that demonstrate their behavior in a 2D plane. Work related to detection of these desired "fingerprints" merits further research. The mathematical description of the DGI approach, tightly associated with a recently published paper [14], is given in the Appendix A.

To complete the first section, Table 1 is provided, describing the existing conventional methods used in electrochemistry together with their shortcomings/limitations.

We want to highlight again that, in comparison with existing methods, the DGI method contains only the *experimental* errors related to direct measurements and the used equipment. The model assumptions and treatment errors are *absent*.

**Table 1.** The basic methods used in electrochemistry for the detection of small additives.

| Method | Basic Limitations | Comments |
|---|---|---|
| Principal component analysis [10] | In single-valued decomposition (SVD), only 3–4 basic components are used. The influence of other components is not evaluated and explained properly. | This is a widely used method in chemometrics for the detection of possible pair correlations between two random sequences. |
| Wavelet decomposition [11] | The criterion of the selection of the proper wavelet from the wide wavelet family is absent. Each chosen wavelet has its own treatment error, and, in many cases, it cannot be evaluated and applied for the analysis of nanonoises in the interval $10^{-9}$–$10^{-6}$ A. | There exist uncontrollable errors related to the application of continuous wavelets to discrete data. This method is important in the detection of small signals associated with the detection of "trace" substances. |
| Timashev's method (flicker-noise spectroscopy) [12]. | The basic supposition is related to a *continuous* representation of an initial sequence (with imposed definitions of intermittency, spikes, outliers, etc.) | The final expressions are derived from the apparatus of continuous mathematics and integral F-transform. These treatment errors are not properly evaluated in analysis of the chosen discrete data. |
| Artificial neural network (ANN) [13] | There is a limited possibility for the physical interpretation of the model parameters. | Analytical possibilities of ANN are stipulated by a wide choice of different transfer functions, and by a wide variation of the number of "neurons" in the intermediate layer, which limits the possibilities of the *optimal* (well-educated) selection of the desired transfer function. |
| Projection on latent structures, partial least squares (PLS) [14] | The PLS method does not allow deriving the reliable calibration model based on the measured voltammograms (VAGs) at multiple sensors because of the appearance of ellipse-like clusters in the score plots. | PLS regression is the generalization of the usual one-dimensional (1D) linear regression for the case of many independent variables. The fitting error cannot be properly evaluated. |

## 2. Materials and Methods

All voltammetric measurements were conducted with the use of potentiostat/galvanostat Elins-P30S (Chernogolovka, Russia) and a three-electrode cell, where the glassy carbon rod was used as a counter electrode and an Ag/AgCl (3.5 M KCl) electrode was used as the reference electrode. The C (graphite) and Pt electrodes were used as the working electrodes.

The added D-tryptophan solution at a concentration of $10^{-3}$ M (SIGMA-ALDRICH, assay $\geq 98.0\%$ (HPLC)) was prepared by dissolving an accurately weighed portion in the background electrolyte. As a background electrolyte, the standard phosphate buffer solution (pH 6.86, a mixture of $Na_2HPO_4$ and $KH_2PO_4$) was used.

In total, 1000 measurements were continuously registered with constant stirring. An amino-acid additive was added after every 100 measurements to the analyzed solution. The concentrations of the analyzed solution after each addition of amino acid are presented in Table 2.

**Table 2.** The values of D-tryptophan added in the given buffer solute (the phosphate buffer solution ($Na_2HPO_4$ + $KH_2PO_4$) with pH 6.86).

| Notation of the Solution Affected by D-Tryptophan Additive | Volume of Additive (mL) | D-Tryptophan Concentration ($10^{-5}$ mol·$L^{-1}$) |
|---|---|---|
| "0"-buffer solution | 0 | 0 |
| "1" | 0.7 | 6.54 |
| "2" | 1.4 | 12.3 |
| "3" | 2.1 | 17.4 |
| "4" | 2.8 | 21.9 |
| "5" | 3.5 | 25.9 |
| "6" | 4.2 | 29.6 |
| "7" | 4.9 | 32.9 |
| "8" | 5.6 | 35.9 |
| "9" | 6.3 | 38.7 |

Each measurement cycle included the following two stages:

(1)    Electrochemical regeneration (precycling)—five successive cycles with a potential scan rate of 2.5 V/s in the range of potentials from 0–2.5 V.
(2)    Registration of the VAG curve in the analyzed solution at a potential scan rate of 0.5 V/s in the range of potentials from 0–1.5 V.

The scheme of the measurement cycle is presented in Figure 1.

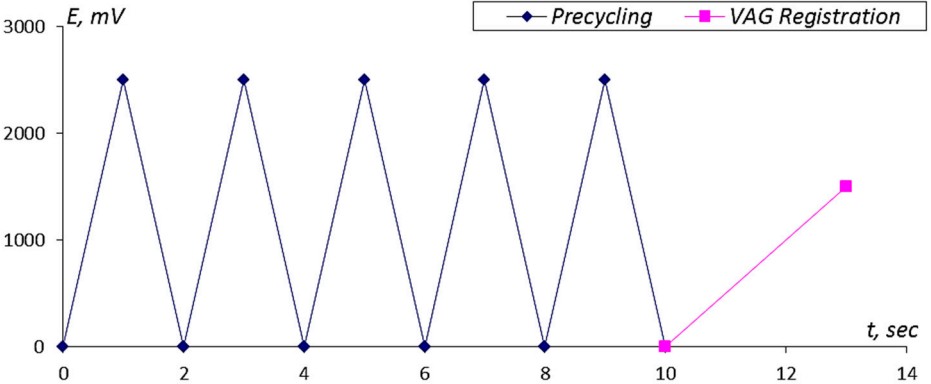

**Figure 1.** The scheme of the measurement cycle. The precycling procedure included five successive cycling stages with a potential scan rate of 2.5 V/s covering the potential range from 0–2.5 V. The voltammogram (VAG) curve of the analyzed solution was registered at a potential scan rate of 0.5 V/s, and the potentials ranged from 0–1.5 V. The time interval between measurements was 10 s. The period of the single VAG registration occupied 3 s.

## 3. Description of the Treatment Procedure

From experimental records, we obtained a set containing 1000 measurements, and the measured set was divided into nine groups; each group, in turn, had 100 measurements (affected by small concentrations of D-tryptophan). The concentration of the added D-tryptophan, the chemical compounds contained in the background buffer solution, and other useful details are listed in Table 2. The initial file with $c = 0$ was determined as the background (content) and other files determined the solution containing the D-tryptophan additives in monotone concentration ($c(x) = \Delta x$, $\Delta = 0.7$, $x = 1, 2,$ . . . , 9; see column 2 in Table 2). The aim of this research was to notice monotone changings evoked by D-tryptophan additives for C- and Pt-electrodes. As an example, we wanted to compare the selectivity of C- and Pt-electrodes to the given additive concentration detected during the extended experiment. However, before proposing an algorithm, we needed to find a definite and justified answer for the following questions:

(a)    Which curve is more sensitive to the presence of a Pt- or C-electrode?
(b)    How many monotone curves can be derived from the parameters that form the invariants of the second/fourth orders?

We actually had two spaces. The data space was formed by the number of data points ($j = 1, 2,$ . . . , $N$). In our case, $N = 1180$. The second space formed the measurement space that incorporated the repeated measurements ($m = 1, 2,$ . . . , $M$) for the given experimental conditions ($M = 100$) that correspond to the relative concentration ($c = 0, 1, 2,$ . . . , 9). The third column of Table 2 gives the true values of $c$. The DGI method enabled finding the answer for the third question.

(c)    Which space is more sensitive to the presence of D-tryptophan in the solute: the data space or the measurement space?

Therefore, we initially prepared the following data that allowed eliminating the influence of the remnant currents. The prepared data satisfied the following requirements:

$$
\begin{aligned}
Yn_m &= \frac{DV_m(U) - \langle DV(U) \rangle_m}{stdev(DV_m(U) - \langle DV(U) \rangle_m)}, \\
DY_m &= Yn_m - \langle Yn \rangle, \quad JY_m = Intergal(U, DY_m), \\
\langle DY \rangle_m &= \frac{1}{N} \sum_{j=1}^{N} DY_{j,m}, \quad \langle F \rangle = \frac{1}{M} \sum_{m=1}^{M} F_m.
\end{aligned}
\tag{1}
$$

Here, $m = 1, 2, \dots, M$ is the number of total measurements in one set ($M = 100$), $DY_m(U)$ defines the initial data file corresponding to the measurement $m$, and the symbol $< \dots >$ determines the arithmetic mean for each fixed measurement in the given space, calculated in accordance with the third line in Equation (1). The number of data points ($j = 1, 2, \dots, N$) for each measurement was equal to $N = 1180$. The symbol $Integral(x, y)$ determines the conventional integral calculated by means of the recurrence trapezoid formulas follows:

$$
Integral(x, y) \Rightarrow J_j = J_{j-1} + \frac{1}{2}(x_j - x_{j-1}) \cdot (y_{j-1} + y_j).
\tag{2}
$$

We should note that $\langle F \rangle = M^{-1} \sum_{m=1}^{M} F_m$ determines the mean function averaged over all measurements. This simple procedure helps essentially eliminate the remnant currents, and it obtains the desired averaged curves that can be used for the comparison of one set of the selected 100 measurements containing additives (with $c = 1, 2, \dots, 9$) with data, corresponding to the buffer solution ($c = 0$). The typical curves corresponding to the comparison of the buffer solution with $c_0 = 0$ and $c_1 = 1$ for C- and Pt-electrodes are given by Figure 2a,b for the normalized data ($dJ/dU$ (derivative) and $J(U)$-usual VAG) in data space. Figure 3a,b demonstrate the behavior of the ranges ($Rg(Y_m)$ and $Rg(JY_m)$) in measurement space for the same concentrations $c_{0,1,9}$ (C-, Pt-electrodes), calculated for the derivatives (Figure 3a) and usual VAGs (Figure 3b). We remind the reader here that the range of a function $f(x)$, defined in the region of a variable $x$ from the interval $[min(x), max(x)]$ and representing itself the maximal deviation is defined as $Rg(f) = max(f) - min(f)$.

This preliminary analysis, supported by Figures 2b and 3b, prompts the choice of the integral curves (usual VAGs) for comparison of the influence of electrodes and the derivation of the desired calibration curves that change monotonically with respect to the applied concentration. We start from the analysis of the parameters that form the invariants of the second order. As shown in the Appendix A, we demonstrate the concentration dependence of the following parameters: the ratio of the gravity centers ($Rt(c) = \langle y \rangle / \langle x \rangle$), the ratio of the quadratic dispersions ($Dsp(c) = \langle DY^2 \rangle / \langle DX^2 \rangle$), parameters $A(c)$ and $B(c)$, defined by Equations (A10) and (A11), and the angle $\alpha(c)$, derived from Equation (A9). It is instructive to realize this comparison for two types of electrodes (C and Pt) and for both spaces (data and measurement). The DGI approach allows using the *same* set of parameters in spite of the fact that the dimensions of two spaces ($N$ and $M$) are different.

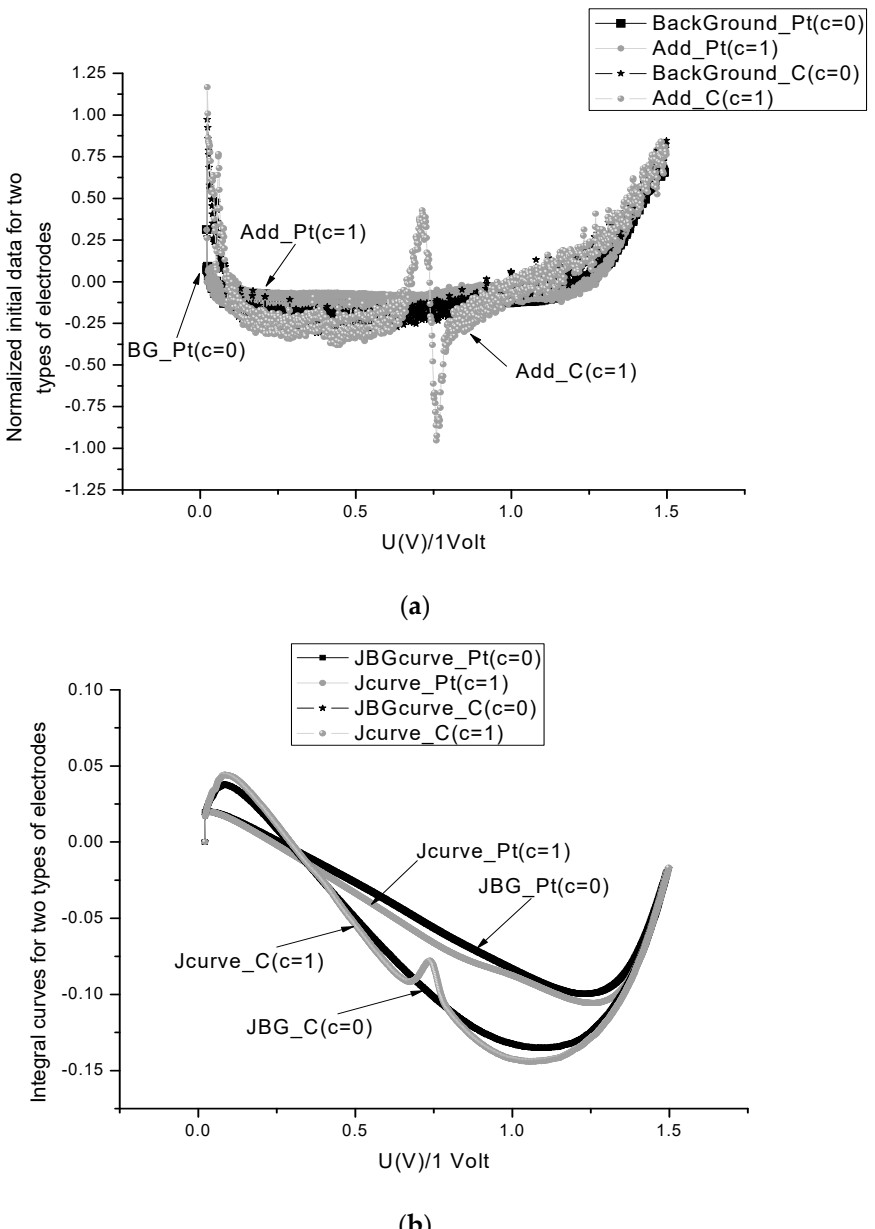

(**a**)

(**b**)

**Figure 2.** (**a**) VAG derivatives (d$J$/d$U$) for two types of electrodes (Pt and C). Minimal concentration ($c$ = 1) corresponds to $6.54 \times 10^{-5}$ mol·L$^{-1}$; see Table 2 for details. (**b**) Normalized and integrated VAGs corresponding to the curves shown in the previous figure. These curves are more preferable for the detection of additives. The integration procedure was realized with the help of Equation (2), which eliminates the additional fluctuations and possible deviations evoked by the presence of an additive. Concentration $c$ = 1 corresponds to the minimal concentration of $6.54 \times 10^{-5}$ mol·L$^{-1}$; see Table 2 for details.

We want to stress here that the standard deviation of each curve, on average, gave 8–10% deviations from the mean curve. We decided *not* to show these error bars for each deviation "tube" in order not to overcharge Figures 4–7 with this excess and inessential information. In the calculation of the invariants of the second and the fourth orders, we chose the mean curves corresponding to the massive dataset compared. Therefore, in all presented figures, the corresponding statistical parameters were obtained for the mean curves. They were calculated with the help of Equation (1), where $M$ = 100.

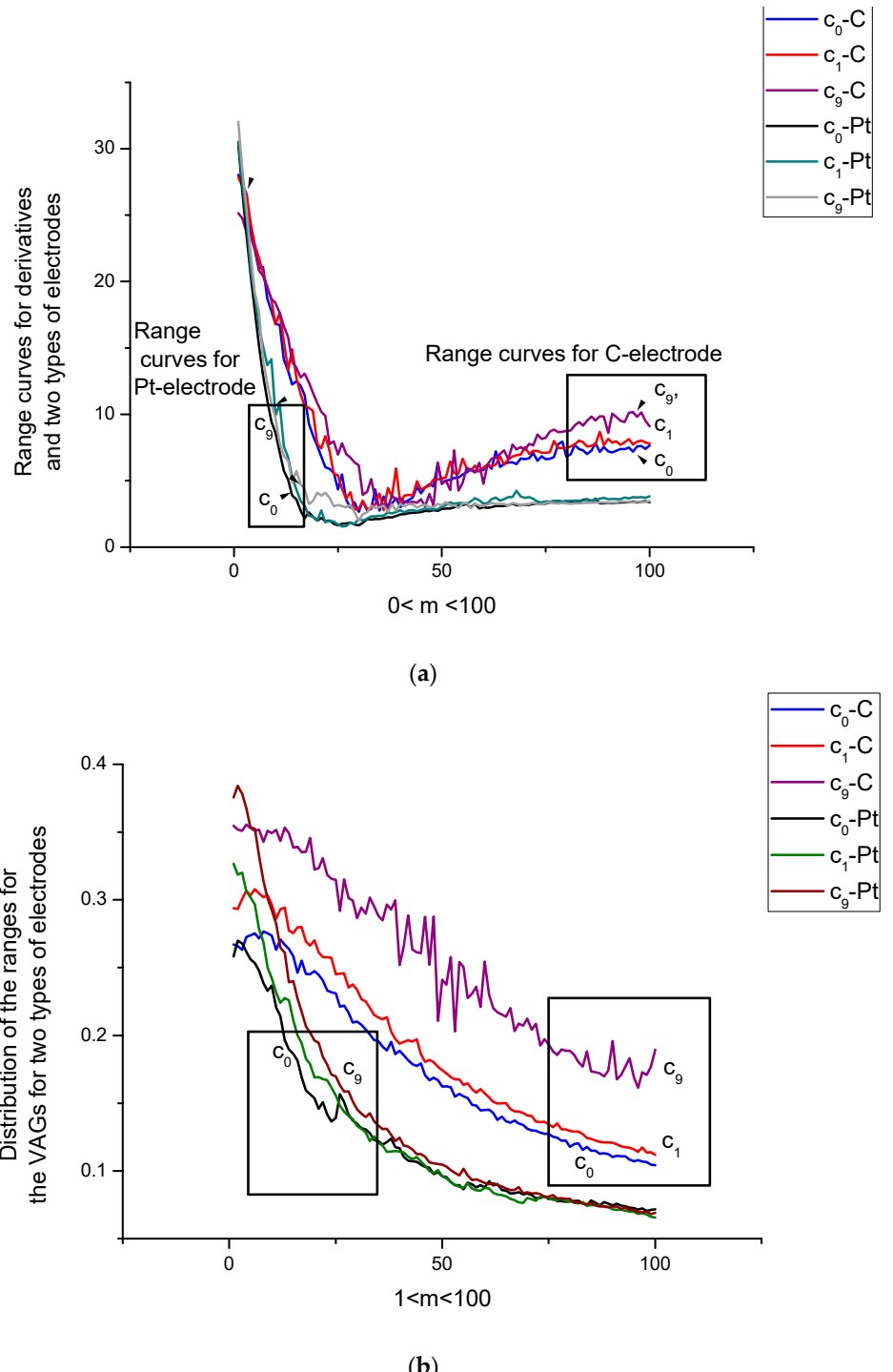

(**a**)

(**b**)

**Figure 3.** (**a**) Distribution of the ranges for differential ($dJ/dU$) VAGs in the measurement space. One hundred measurements for $c_{0,1,9}$ were repeated in the same experimental (temperature, pressure, humidity, pH) conditions. The period of one VAG registration was 3 s. The duration between measurements was 10 s. One can notice that the chosen minimal concentration $c_1$ was very close to the curve, corresponding to the $c_0$ concentration for both types of electrodes. All measurements were reduced to the same interval [1,100]. (**b**) Distribution of the ranges for the usual VAGs for the same values of concentrations $c_{0,1,9}$. In comparison with the previous figure, these curves become more sensitive to the presence of D-tryptophan in the buffer solute and, hence, more preferable for further analysis.

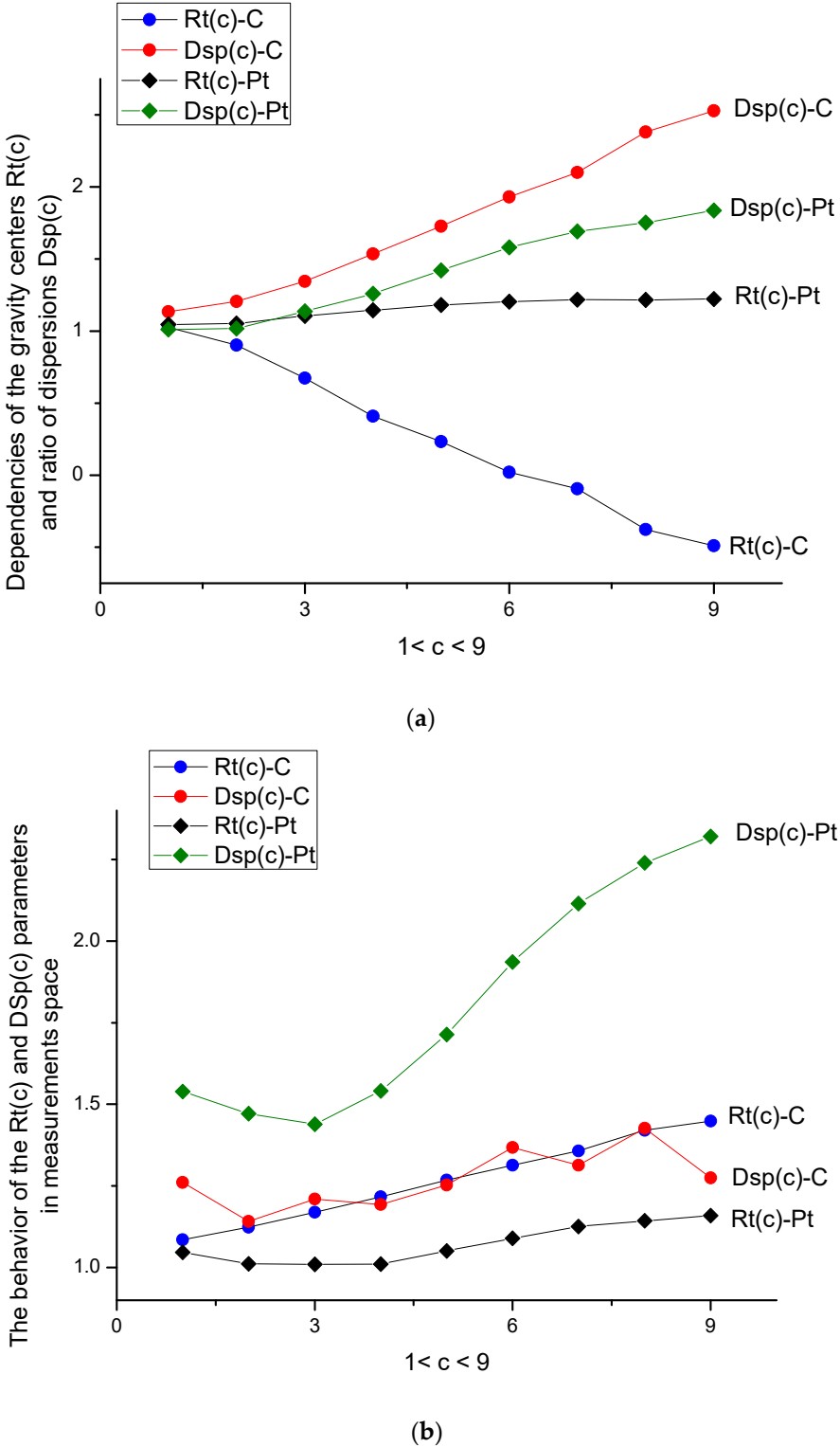

**Figure 4.** (**a**) Behavior of parameters Rt(*c*) = <y>/<x> and Dsp(*c*) = <D$Y^2$>/<D$X^2$> for two types of electrodes in data space. All these curves changed monotonically with respect to the increase in D-tryptophan concentration. The true c-values are given in Table 2. (**b**) Behavior of the same parameters Rt(*c*) = <y>/<x> and Dsp(*c*) = <D$Y^2$>/<D$X^2$> is shown for two types of electrodes in the measurement space. Only two curves of Rt(*c*) for two types of electrodes kept their monotone behavior, while the two other curves for dispersion ratios lost (completely or partly) their monotone behavior.

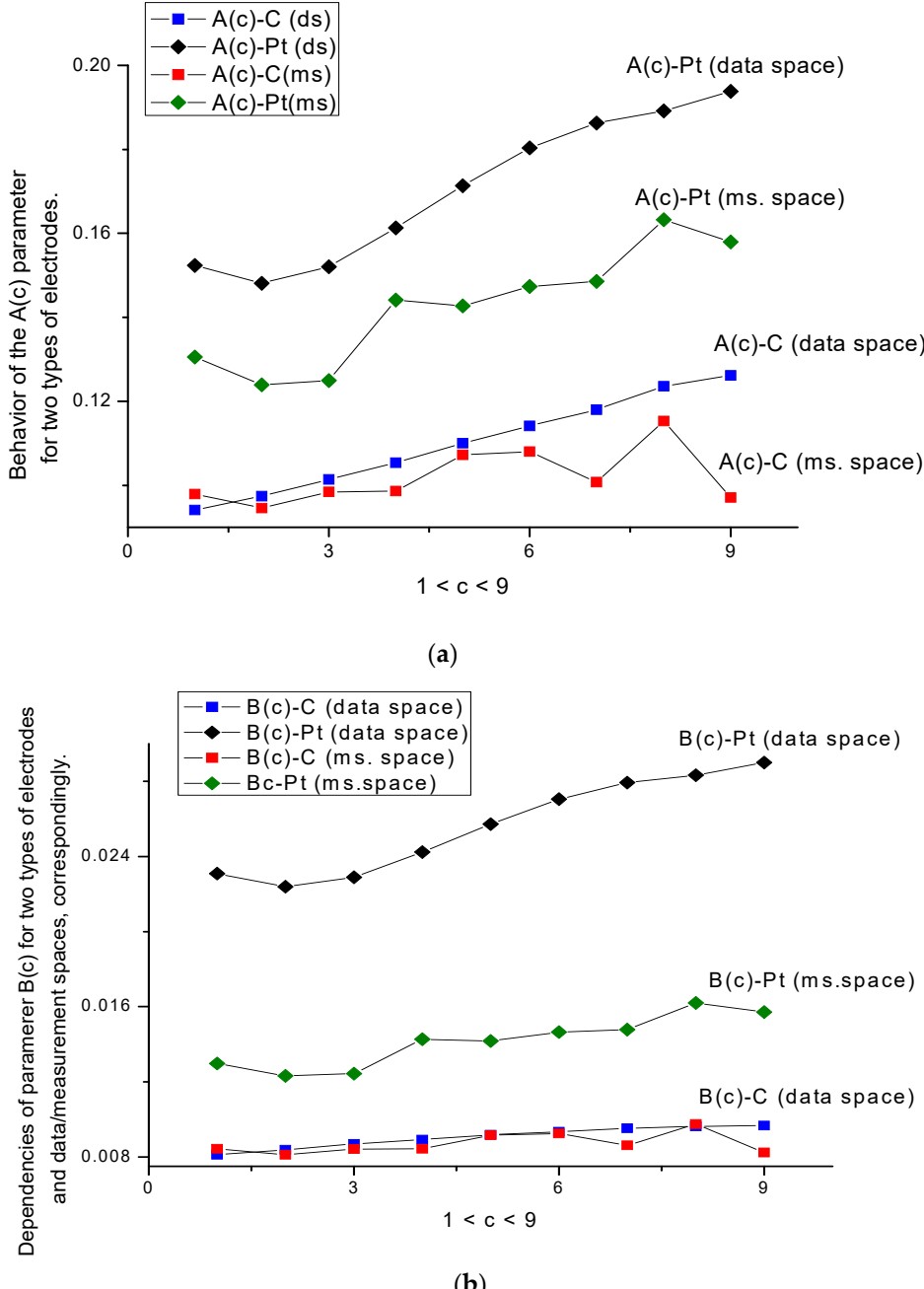

**Figure 5.** (**a**,**b**) Behavior of parameters *A*(*c*) and B(c) for two types of electrodes combined for the data and measurement spaces. The scale of these parameters allows this. The monotone behavior of the *A*(*c*) and B(c) curves for the C-electrode in data space is clearly expressed, while, for the Pt-electrode, the calculated curves in data space kept their monotone behavior only in part. In the measurement space, four curves *A*(*c*) and B(c) for both types of electrodes completely lost their monotone behavior.

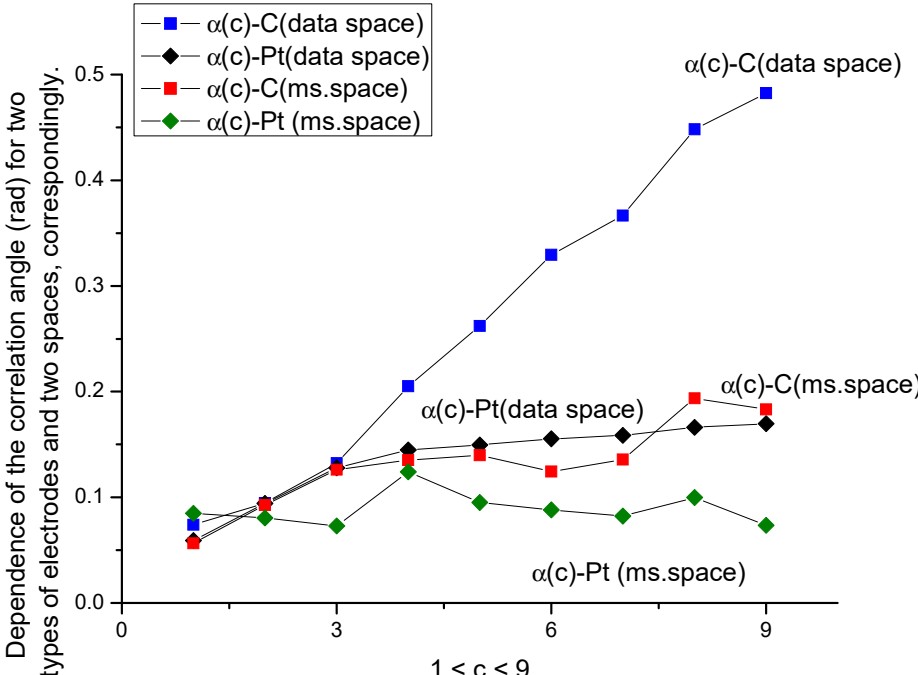

**Figure 6.** Behavior of the correlation angle $\alpha(c)$ for two types of electrodes in data and measurement spaces. We observe the monotone behavior for C- and Pt-electrodes in data space, while, for the measurement space, these curves for both types of electrode completely lost their monotone behavior.

These figures finalize the preliminary analysis realized for the main parameters characterizing the behavior of DGI of the second order. Coming back to the analysis of the parameters that characterize the behavior of the invariant of the fourth order (see Equations (A16) and A(17)), we should mark the following:

1.  We should take into account the behavior of parameter $\sigma_{B,C}(c)$, which incorporates the influence of the higher moments. This dependence can be important for other applications.
2.  The behavior of the value of the invariant $I_4(c)$ (derived from Equation (A17))) is also important and should be taken into account, as well.

In order to avoid the duplicity of figures, we do not show the dependencies of the parameter $S_{A,B,C}(c)$ (defined by Equation (A16)), because the exhibited behavior repeats the behavior of parameters $A(c)$ and $B(c)$, considered above. We omit also the dependencies of curves from Equations (A7) and (A18) against the polar angle $\varphi$, because their forms are not important here for further analysis. However, we want to stress also the fact that the curve from Equation (A18) is more informative, because it can have both a real part $Re(x(\varphi))$, $Re(y(\varphi))$ and a complex conjugated part $Im(x(\varphi)-<x>))$, $Im(y(\varphi)-<y>)$.

More deep analysis related to these figures is discussed in the final section.

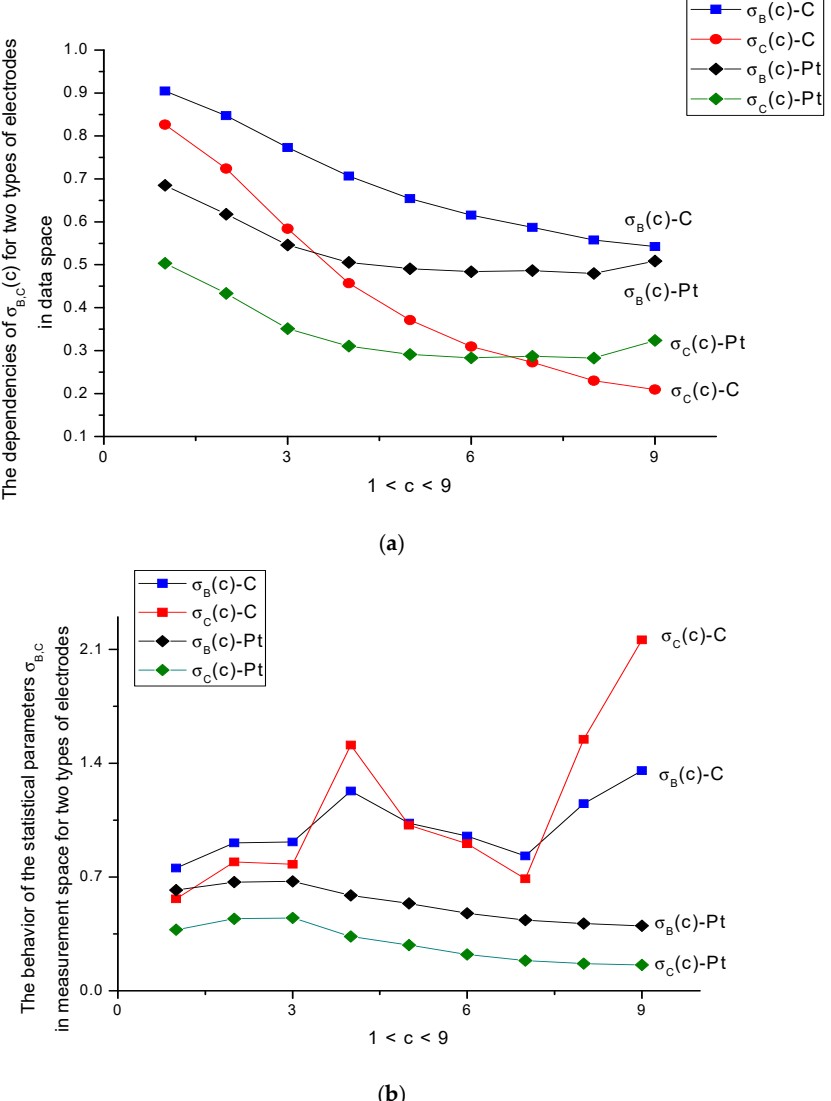

(**a**)

(**b**)

**Figure 7.** (**a**) Monotone behavior of the statistical parameters $\sigma_B(c)$ and $\sigma_C(c)$ in data space that can be chosen for calibration purposes. They independently confirm the behavior of the ellipse rotation (correlation) angle (see Figure 6) found for the discrete geometrical invariants (DGI) of the second order. (**b**) Random-like (non-monotone) behavior of the statistical parameters $\sigma_B(c)$ and $\sigma_C(c)$ in the measurement space for the C-electrode, which cannot be chosen for calibration purposes. They reflect the strong fluctuation characteristic of the measurement space, confirmed also by the parameters characterizing the invariant of the second order. As for the Pt-electrodes, these curves demonstrate quasi-monotone behavior.

## 4. Results and Discussions

Voltammetric behavior of C- and Pt-electrodes in the absence of an electroactive substance was described earlier [15]. In Reference [15], we gave the results of decoding a voltammetric series with the help of the DGI method. In the practical aspect, it is interesting to compare the results presented in this paper with results obtained earlier. The special interest is related to parameters calculated for the integral voltammetric curves. In the presence/absence of electroactive substance in the repeated sensor, the integral curves allow obtaining more monotone parameters in data space, which reflects the modification of the sensor surface and variations of the solution content at small variations of the electroactive component's concentration. It is interesting to note that the results obtained earlier for the background solution allow quantitatively characterizing the aging phenomenon;

the repeated cycles of electrochemical oxidation/reduction of the components of the background electrolyte are accompanied by surface electrode degradation. This phenomenon was observed for C- and Pt-electrodes. In voltammetry, there is a signal drift problem and a sensor memory effect tightly related to it. The elimination of this negative effect is impossible, especially for cases where a large number of electrodes are used to form the given multisensor system. One solution is to use special mathematical methods for a quantitative description of the random noises and signal drifts aggravated by the measured equipment interferences. Comparing the preliminary results obtained earlier in Reference [15], one can formulate the following conclusions:

(1) The C- and Pt-electrodes at the repeated functioning cycles exhibit the aging phenomenon that leads, in turn, to the temporal signal drift. This drift distorts the measured VAGs. This effect is expressed clearly in the analysis of the curves associated with the measurement space. For a quantitative description, as well as the control and elimination of this negative phenomenon, we use the DGI method that frees the system from the imposed model and errors related to the treatment procedure.

(2) The integral curves in the measurement space because of the DGI application demonstrate the sensitivity of both electrodes to the amino-acid concentration variations (see the monotone curves obtained for the data space).

(3) The aging effect confirms our previous conclusions; after achieving the activated state, the electrode behavior at the given potential sweep depends on the number of cycles to a lesser degree. This effect is expressed clearly for the more inertial Pt-electrodes. For C-electrodes, we presumably observe the varied behavior covering all 1000 measurements.

(4) The ratio $Rt(c) = <y(c)>/<x>$ (remember that the variable $<x> = <y(0)>$ is always related to the buffer solution), while dispersion $Dsp(c) = <DY^2>/<DX^2>$, depicted in Figure 4a, characterizes the center of gravity and its standard deviation. They are more sensitive to the surface changes evoked by the aging phenomenon. Both parameters have monotone behavior for C- and Pt-electrodes in the presence/absence of the amino acid. For the Pt-electrode, this monotone dependence is weak; it confirms the high stability of the Pt-electrode during electro oxidation processes. We should stress here that Figure 4a,b are more informative and allow characterizing the D-tryptophan behavior, as well as comparing its electroactivity with the interaction of C- and Pt-electrodes.

    (a) In data space, the dispersion curves Dsp for C- and Pt-electrodes differ by their mutual location with respect to each other; the value $Dsp(c)$ (C-electrode) exceeds the curve $Dsp(c)$ (Pt-electrode) for all values of *c*. The gravity ratios $Rt(c)$ for both types of electrodes are strongly deviated from each other as well.

    (b) In measurement space, a significant difference is observed for the curve Dsp(c) (Pt). It is explained by the excess electroactivity of the tryptophan molecule in its interaction with the Pt-electrode. One can stress here that the clearly expressed peaks on the initial VAGs were not observed.

Therefore, one can conclude that parameters $Rt(c)$ and $DSp(c)$ contain qualitative information about the nature of the depolarizator and the corresponding electrode.

(5) The behavior of parameters $A(c)$ and $B(c)$ (Figure 5a,b) is more sensitive to the surface changes; the behavior of parameter $A(c)$ for both types of electrodes is characterized by a monotone behavior, which increases in the presence/absence of the amino acid.

(6) The ellipse rotation (correlation) angle $\alpha$ for the C-electrode in the absence of amino acid does not correlate with the number of oxidation/reduction cycles; furthermore, it does not depend on the sensor duration and cannot be associated with the aging phenomenon. However, as shown in Figure 6, this parameter correlates strongly with D-tryptophan concentration. The calculated

curve is monotone with respect to increasing the concentration of D-tryptophan. For the Pt-electrode after 300 cycles in the presence/absence of amino acid, this curve has an increasing trend. We relate this increasing tendency with additional processes of oxidation in the background solution. From our point of view, this parameter can be used for calibration purposes of some non-electroactive components of the background solute.

(7) The parameters analyzed above reflect the specificity of electrode processes from different points of view. For the more active C-electrode, which has better oxidation characteristics, many output DGI parameters changed more intensively in comparison with the more inertial Pt-electrode. The Pt-electrode is more stable in oxidation processes. Being a more "stable" material with respect to the transfer electron process related to the oxidation of amino acids, it requires the calculation of an extended set of "fine" parameters that are sensitive to concentration variations of the electroactive substance. For the given case, we associate these variations with the behavior of parameter $\sigma_{B,C}$. As it follows from the analysis of Figure 7a, these parameters have monotone and decreasing characteristics against the concentration in data space. In the measurement space shown in Figure 7b, these parameters characterize again the electroactivity of the C-electrode, which has variable characteristics, compared to the relatively stable behavior of the Pt-electrode. Again, the 300-cycle boundary is conserved. Only after this limit does the Pt-electrode tend to the aging regime.

(8) The proposed DGI approach is rather general and enables comparing random curves with different samples, using the *same* set of parameters expressed in terms of moments up to the fourth order. We showed also that these parameters have different sensitivity with respect to the input factor (in our case, this factor is associated with concentration of the amino acid). Comparing both spaces, one can say that the data space is more stable in comparison with the measurement space; the latter actually demonstrates the temporal evolution (each measurement together with "idle" time occupies 13 s) of the whole process of the interaction of the amino acid with the given electrode. The DGI approach allows obtaining a set of calibration curves and establishing a natural limit associated with parameter $I_4(c)$ (see Figure 8 for details). This parameter has a small value and loses its monotone behavior for the both spaces.

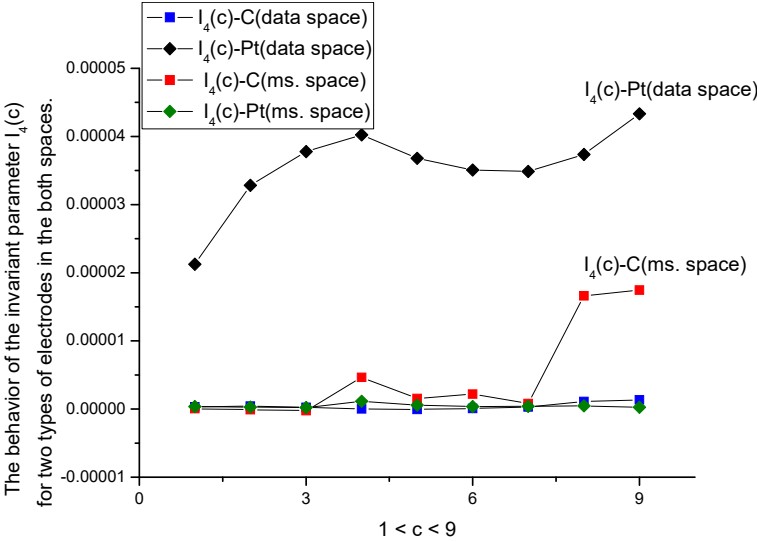

**Figure 8.** Behavior of the parameter $I_4(c)$ from Equation (A17). Having small values located in the interval $10^{-6}$ to $5 \times 10^{-5}$, it has a maximal sensitivity to the presence of an uncontrollable factor that can influence the interaction between the chosen electrode with D-tryptophan additive. The two curves below $I_4(c)$-C (in data space) and $I_4(c)$-Pt (in measurement space) do not have monotone characteristics. They are located in the interval $10^{-9}$ to $10^{-6}$. Therefore, they appear as a couple of horizontal lines with values close to zero.

**Author Contributions:** Conceptualization, R.N. and H.B.; methodology, R.N. and A.S.; software, E.M., A.V.; validation, R.N., H.B. and A.S.; data curation, A.S., E.M.; writing—original draft preparation, R.N.; writing—H.B., A.S. and R.N.; editing, R.N.; visualization, R.N.; supervision, H.B.; project administration, A.S.; funding acquisition, A.S.

**Funding:** This work was partly supported by the Russian Foundation for Basic Research, project No. 17-43-020232 r-Povolzh'ye-a.

**Conflicts of Interest:** The authors declare no conflict of interest.

**The Basic Abbreviations**

DGI     Discrete geometrical invariant
SVD     Single-valued decomposition
VAG(s)  Voltammogram(s)

**Appendix A**

In this Appendix, we reproduce the basic ideas of the DGI approach and necessary formulas that are necessary for understanding the proposed algorithm outlined in Section 3.

In the book of Babenko [16], it was shown that the well-known Pythagor's theorem can be *generalized* and propagated for a set of *random* points having *discrete* coordinates $(x_k, y_k)$ ($k = 1, 2, 3, \ldots, n$). Let us consider the square of the distance connecting an arbitrary point $M(x, y)$ with the $k$th point $(x_k, y_k)$ belonging to two given sets.

$$l_k^2 = (x - x_k)^2 + (y - y_k)^2. \tag{A1}$$

We require that

$$\frac{1}{n} \sum_{k=1}^{n} l_k^2 = I^2 \equiv const. \tag{A2}$$

Inserting Equation (A1) into (A2), we obtain

$$
\begin{aligned}
&(x - \langle x \rangle)^2 + (y - \langle y \rangle)^2 = I^2 - R^2, \\
&\langle x^p \rangle = \frac{1}{n} \sum_{k=1}^{n} x_k^p, \ \langle y^p \rangle = \frac{1}{n} \sum_{k=1}^{n} y_k^p, \ R^2 = \langle \Delta x^2 \rangle + \langle \Delta y^2 \rangle, \\
&\langle \Delta V^2 \rangle = \langle V^2 \rangle - \langle V \rangle^2.
\end{aligned}
\tag{A3}
$$

As one can notice from Equation (A3) that the set of circles can exist if the invariant $I^2 \geq R^2$, where the equality sign corresponds to a circle with zero radius; it is convenient to consider the invariant circle with radius $I^2 = 2R^2$. From another point of view, the requirement of Equation (A2) can be considered as the reduction of the given set of points to the continuous circle with four statistical parameters ($<x^p>$, $<y^p>$; $p = 1, 2$).

However, for practical purposes this simplest requirement in Equation (A2) is *not* sufficient and, therefore, it makes sense to consider other geometrical combinations. In order to reduce the curve with eight statistical parameters, we consider another combination that is more complex in comparison with Equation (A1).

$$
\begin{aligned}
&L_k^{(2)} = C^2 (y - y_k)^2 - 2B(x - x_k) \cdot (y - y_k) + A^2 (x - x_k)^2, \\
&k = 1, 2, \ldots, n
\end{aligned}
\tag{A4}
$$

The quadratic form in Equation (A4) contains five statistical parameters ($<x^p>$, $<y^p>$, $p = 1, 2$; $<xy>$) and three unknown parameters ($A$, $B$, $C$). We subordinate this combination to the following requirement:

$$\frac{1}{n} \sum_{k=1}^{n} L_k^{(2)} = I^2 \equiv const. \tag{A5}$$

Inserting Equation (A4) into (A5), after simple algebraic manipulations, one can obtain

$$C^2(y - \langle y \rangle)^2 - 2B(y - \langle y \rangle) \cdot (x - \langle x \rangle) + A^2(x - \langle x \rangle)^2 + E^2 \equiv I^2,$$
$$E^2 = C^2 \langle \Delta y^2 \rangle - 2B \langle \Delta x \Delta y \rangle + A^2 \langle \Delta x^2 \rangle. \tag{A6}$$

As before, we put $I^2 = 2E^2$. In order to find three unknown parameters ($A$, $B$, $C$), it is convenient to use the obvious parameterization for the variables ($x$, $y$) relative to the angle $\varphi$.

$$y = \langle y \rangle + A \cos(\varphi - \alpha),$$
$$x = \langle x \rangle + C \cos(\varphi), \ 0 \leq \varphi \leq 2\pi. \tag{A7}$$

Excluding the parameter $\varphi$ from Equation (A7) and identifying expression (A6) with expression (A8),

$$C^2 \left\langle (\Delta y)^2 \right\rangle - 2AC \cos \alpha \langle (\Delta x) \cdot (\Delta y) \rangle + A^2 \left\langle (\Delta x)^2 \right\rangle \equiv A^2 C^2 - B^2,$$
$$E^2 = C^2 \langle \Delta y^2 \rangle - 2AC \cos \alpha \langle \Delta x \Delta y \rangle + A^2 \langle \Delta x^2 \rangle = A^2 C^2 \sin^2 \alpha, \tag{A8}$$

we obtain

$$\cos \alpha = \frac{B}{AC}, \ E^2 = A^2 C^2 - B^2. \tag{A9}$$

In order to decrease the number of unknown parameters, we find from Equation (A7) the values of $A$ and $C$ from the obvious conditions.

$$y_{max} = \langle y \rangle + A, \ y_{min} = \langle y \rangle - A, \ \rightarrow A = \tfrac{1}{2}(y_{max} - y_{min}),$$
$$x_{max} = \langle x \rangle + C, \ x_{min} = \langle x \rangle - C, \ \rightarrow C = \tfrac{1}{2}(x_{max} - x_{min}). \tag{A10}$$

Parameter $B$ is found from relationships in Equations (A8) and (A9) as a positive root of the quadratic equation written as follows.

$$B^2 - 2\langle \Delta x \Delta y \rangle B - \left[ A^2 C^2 - \langle \Delta x^2 \rangle A^2 - \langle \Delta y^2 \rangle C^2 \right] = 0,$$
$$B = \langle \Delta x \Delta y \rangle + \left[ (\langle \Delta x \Delta y \rangle)^2 + A^2 C^2 - \langle \Delta x^2 \rangle A^2 - \langle \Delta y^2 \rangle C^2 \right]^{1/2}. \tag{A11}$$

This root is chosen from the comparison of two identity sequences ($x_k = y_k$) that follows from the obvious requirement $B = A^2$, ($\alpha = 0$). Concluding this section, one can say that, with the help of the rotated counterclockwise ellipse in Equation (A7), we reduced $2n$ random points figuring in equation (A4) to eight statistical parameters $\text{Pr}_8^{(2)}$: ($<x^p>$, $<y^p>$, $p = 1,2$; $<xy>$, $A$, $C$, $\alpha$). Equations (A7)–(A9) can be considered as the geometrical interpretation of the conventional Pearson correlation coefficient defined by Equation (A9).

In order to increase the sensitivity in the detection of the hidden distortions that are contained in two random curves, one can increase the order of the calculated moments. For this purpose, one can consider the invariant of the fourth order.

$$L_k^{(4)} = A_4(x - x_k)^4 + 2B_4(x - x_k)^2(y - y_k)^2 + C_4(y - y_k)^4. \tag{A12}$$

Inserting Equation (A12) into Equation (A13),

$$\frac{1}{n} \sum_{k=1}^{n} L_k^{(4)} = Inv, \tag{A13}$$

and equating the linear terms,

$$X \equiv x - \langle x \rangle, \ Y \equiv y - \langle y \rangle, \tag{A14}$$

to zero, one can obtain the following combination:

$$
\begin{aligned}
K(X,Y) &= K_2(X,Y) + K_4(X,Y) = I_4. \\
K_2(X,Y) &= A_2 X^2 - 2B_2 X \cdot Y + C_2 Y^2, \\
K_4(X,Y) &= A_4 X^4 - 2B_4 X^2 Y^2 + C_4 Y^4, \\
I_4 &= A_4 \left\langle (\Delta x)^4 \right\rangle - 2B_4 \left\langle (\Delta x)^2 (\Delta y)^2 \right\rangle + C_4 \left\langle (\Delta y)^4 \right\rangle.
\end{aligned}
\tag{A15}
$$

The combinations shown below define the constants figuring in the DGI (Equation (A15)).

$$
\begin{aligned}
B_4 &= \frac{\left\langle (\Delta x)^3 \right\rangle}{\left\langle (\Delta x)(\Delta y)^2 \right\rangle} A_4 \equiv \sigma_B A_4, \\
C_4 &= \frac{\left\langle (\Delta x)^3 \right\rangle}{\left\langle (\Delta y)^3 \right\rangle} \frac{\left\langle (\Delta y)(\Delta x)^2 \right\rangle}{\left\langle (\Delta x)(\Delta y)^2 \right\rangle} A_4 \equiv \sigma_C A_4, \\
A_2 &= \left[ 6 \left\langle (\Delta x)^2 \right\rangle - 2\sigma_B \left\langle (\Delta y)^2 \right\rangle \right] A_4 \equiv S_A A_4, \\
B_2 &= 4\sigma_B \langle \Delta x \Delta y \rangle A_4 \equiv S_B A_4, \\
C_2 &= \left[ 6\sigma_C \left\langle (\Delta y)^2 \right\rangle - 2\sigma_B \left\langle (\Delta x)^2 \right\rangle \right] A_4 \equiv S_C A_4.
\end{aligned}
\tag{A16}
$$

The constant $I_4$ and *Inv* from Equation (A13) are defined by Equation (A17)

$$
Inv = 2I_4 = 2\left[ \left\langle (\Delta x)^4 \right\rangle - 2\sigma_B \left\langle (\Delta x)^2 (\Delta y)^2 \right\rangle + \sigma_C \left\langle (\Delta y)^4 \right\rangle \right] A_4.
\tag{A17}
$$

The averaged values $<(\Delta x)^q (\Delta y)^p>$ characterizing two compared sets are defined by Equation (A13). The polynomial $K(X,Y)$ of the fourth order can be separated in the polar coordinate system. Using the notations in Equation (A16) and taking into account the fact that the constant $A_4$ figuring in $K(X,Y)$ is an arbitrary proportion multiplier and, therefore, can be omitted, we present the desired curve in the form

$$
\begin{aligned}
x(\varphi) &= \langle x \rangle + r(\varphi) \cos \varphi, \\
y(\varphi) &= \langle y \rangle + r(\varphi) \sin \varphi, \\
r(\varphi) &= \left[ \frac{\sqrt{q_2^2(\varphi) + 4I_4 q_4(\varphi)} - q_2(\varphi)}{2q_4(\varphi)} \right]^{1/2}.
\end{aligned}
\tag{A18}
$$

The functions $q_{2,4}(\varphi)$ figuring in Equation (A18) are determined as follows:

$$
\begin{aligned}
q_2(\varphi) &= S_A \cos^2(\varphi) - 2S_B \sin \varphi \cos \varphi + S_C \sin^2(\varphi), \\
q_4(\varphi) &= \cos^4(\varphi) - 2\sigma_B \sin^2 \varphi \cos^2 \varphi + \sigma_C \sin^4(\varphi).
\end{aligned}
\tag{A19}
$$

This curve of the fourth order containing again eight statistical parameters $\mathrm{Pr}_8^{(4)}$ ($<x>$, $<y>$, $\sigma_B$, $\sigma_C$, $S_{A,B,C}$, $I_4$) determines the statistical proximity/difference of 2D random curves/sets located in the plane. What happens if two random curves are identical to each other ($x_j = y_j$) for all numbers of the discrete points $j = 1, 2, \ldots, N$? In this case, as it is easy to see from Equation (A18), $\sigma_B = \sigma_C = 1$, $I_4 = 0$, $S_{A,B,C} = 4 <(\Delta x)^2>$ and, therefore, from Equation (A18), it follows that $r(\varphi) = 0$. In this case, Equation (A18) is degenerated into a *point* with coordinates $<x> = <y>$, located on the line $y = x$.

We want to stress here that the algorithm proposed in Reference [15] is *original* and differs from the mathematical proofs that are given in Reference [16]. In this paper, we consider Equations (A7) and (A18) and parameters $\mathrm{Pr}_m^{(2)}$, $\mathrm{Pr}_m^{(4)}$ ($m = 1, 2, \ldots, 8$) that reflect the different sensitivities with respect to a chemical additive presented in the measured VAG(s).

In conclusion, we want to stress that the term DGI reflects the discrete structure ($x_k$, $y_k$) ($k = 1, 2, \ldots, n$) of the compared random sets, their descriptive and geometric forms in the 2D plane, and their requirement of the constant *Inv* in each specific form. The consideration of other DGIs containing the moments of the fifth, sixth, etc. orders becomes *problematic*, because these invariants do not allow realizing the separation procedure in a polar coordinate system.

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
