# Peer review of "Detection of Additives with the Help of Discrete Geometrical Invariants"

_applsci, doi:10.3390/app9050926_

Round 1
Reviewer 1 Report
Rephrase/Grammar
#42-42 : Rephrase for clarity.
#82-84: "..becoming useless"
#105-106
#108: stressed (past tense?)?
#218, 226-227: correspondingly
#219-227: combining ?
#221, 229: with regards..
Fig. 5(a) and 5 (b); 7(a), 7(b) the descriptions should be combined together.
Captions should be re-written for clarity and brevity.
Introduction
DGI: Better introduction for the general readers.
Why D-tryptophan?
Explain/Reason
#84: Inertial behavior
#86: differed behavior?
#117: background?
MAJOR
Experiment/Results
Missing error bars.
Must be shown in all the presented results.
Reviewer 2 Report
The changes made to the paper greatly improve its readability and understanding. The data is better organized and more easily accessible.
Moving the mathematical equations to the appendix greatly facilitates reading.
Schemes have also been improved.
The paper has greatly improved.
English language is excellent.
Reviewer 3 Report
Reviewer report on Manuscript Draft entitled ‘Detection of the Additives with the Help of the Discrete Geometrical Invariants’.
In this work authors propose a mathematical method for the detection of electrochemical additives in solution using Discrete Geometrical Invariants (DGI). Authors claims that the results presented in this paper opens a new direction for solution of the actual problem, which exists in analytical chemistry, i.e. detection of small additives in the given solute and their further description using monotone/calibration curve expressed by means of parameters associated with the found DGI. The manuscript at some extent probably could be interesting from methodological point of view.
However, despite of some attempts, theoretical assumptions are not very clearly and not very well supported by experimental data, therefore, I am recommending the rejection of this manuscript.
Corrections were not indicated clearly.
Some minor points are:
Abstract starts like Introduction part even with citation references, what is not very usual, because abstract will be represented in various different media, which does not have option to include references. Therefore, it is better to cite references in Introduction and/or other consecutive parts of manuscript.
Legend of Figure 1 is not informative enough, it will be nice to present how electrode was pre-treated and how analytical signal was generated, not clear how the signal was generated (what was added?). It will be nice to have this information in legend.
Some other legends of figures contains information, which is more suitable for discussion part (e.g. Fig. 6, 7).
English needs improvements.
Round 2
Reviewer 1 Report
#258-260: It is not clear if the mean curves are presented. The captions should mention this as well number of cycles.
Reviewer 3 Report
Some corrections were performed, now manuscript is suitable for publishing.
